# Treatment of Mouse Infants with Amoxicillin, but Not the Human Milk-Derived Antimicrobial HAMLET, Impairs Lung Th17 Responses

**DOI:** 10.3390/antibiotics12020423

**Published:** 2023-02-20

**Authors:** Sudhanshu Shekhar, Navdeep Kaur Brar, Anders P. Håkansson, Fernanda Cristina Petersen

**Affiliations:** 1Institute of Oral Biology, University of Oslo, 0316 Oslo, Norway; 2Division of Experimental Infection Medicine, Department of Translational Medicine, Lund University, 21428 Malmö, Sweden

**Keywords:** amoxicillin, HAMLET, infants, lungs, Th17 immunity

## Abstract

Emerging evidence suggests differential effects of therapeutic antibiotics on infant T cell responses to pathogens. In this study, we explored the impact of the treatment of mouse infants with amoxicillin and the human milk-derived antimicrobial HAMLET (human alpha-lactalbumin made lethal to tumor cells) on T cell responses to *Streptococcus pneumoniae*. Lung cells and splenocytes were isolated from the infant mice subjected to intranasal administration of amoxicillin, HAMLET, or a combination of HAMLET and amoxicillin, and cultured with *S. pneumoniae* to measure T cell responses. After *in-vitro* stimulation with *S*. *pneumoniae*, lung cells from amoxicillin- or amoxicillin plus HAMLET-treated mice produced lower levels of Th17 (IL-17A), but not Th1 (IFN-γ), cytokine than mice receiving HAMLET or PBS. IL-17A/IFN-γ cytokine levels produced by the stimulated splenocytes, on the other hand, revealed no significant difference among treatment groups. Further analysis of T cell cytokine profiles by flow cytometry showed that lung CD4+, but not CD8+, T cells from amoxicillin- or HAMLET plus amoxicillin-treated mice expressed decreased levels of IL-17A compared to those from HAMLET-exposed or control mice. Collectively, these results indicate that exposure of infant mice to amoxicillin, but not HAMLET, may suppress lung Th17 responses to *S. pneumoniae*.

## 1. Introduction

Antibiotics are among the most commonly used drugs for neonates and infants that are suffering from or are prone to bacterial infections, particularly sepsis, which causes severe morbidity and mortality across the globe [1,2,3,4]. Among newborns, neonatal sepsis causes an estimated 600,000 deaths per annum worldwide [5]. Amoxicillin, a β-lactam penicillinase-susceptible semisynthetic amino-penicillin antibiotic with activity against a wide range of bacteria, is one of the most commonly prescribed antibiotics for the treatment or prevention of neonatal sepsis [6,7]. Although antibiotics are desired to specifically target pathogenic bacteria, many antibiotics, including amoxicillin, have reported adverse side effects on the neonatal and infant microbiota and immunity that can contribute to the development of dysbiosis, microbiota perturbance, and impaired immunity to pathogens [8,9]. These side effects are much more profound and long-lasting in neonates and infants due to their evolving and immature immunophysiological systems [8,9,10].

There are a limited number of studies that focus on the impact of antibiotic regimens on neonatal and infant T cell responses to pathogens [8,9,10,11,12,13,14]. Gonzalez-Perez et al. demonstrated that perinatal exposure of mice to a combination of ampicillin, streptomycin, and clindamycin after vaccinia virus infection reduced the number of virus-specific neonatal/infant CD8+ T cells expressing IFN-γ [11]. Not only that, but the infants showed altered peripheral CD8+ T cell receptor signaling due to the gastrointestinal microbiome dysbiosis [12]. In accordance with these suppressive effects on T cell function, we recently showed that upon in vitro stimulation with *Streptococcus pneumoniae*, CD4+, but not CD8+, T cells from neonatal mice exposed to piperacillin in combination with the beta-lactamase inhibitor tazobactam expressed lower levels of IL-17A (Th17) and IFN-γ (Th1) cytokines compared to unexposed mice [13]. On the other hand, when newborn pigs exposed to therapeutic amoxicillin doses were challenged with *Salmonella enterica* serovar Typhimurium, their whole blood analysis exhibited an enhanced upregulation of effector T cell surrogate cytokine, IFN-γ, gene expression compared to those treated with placebo [15]. Similarly, CD4+ T cells from antibiotic-exposed (ampicillin plus neomycin) infant mice immunized with the antipneumococcal conjugate vaccine PCV13 showed increased IFN-γ recall responses [16]. Altogether, while these data have shed some light on the differential effects (stimulatory versus suppressive) of different antibiotic types on T cell immunity, the impact of amoxicillin on T cell responses to pathogens, despite being routinely administered to neonates and infants in clinical settings, is largely unknown.

HAMLET (human alpha-lactalbumin made lethal to tumor cells) is a human milk-derived lipid-protein complex that possesses bactericidal activities against certain Gram-positive and Gram-negative bacteria [17,18,19]. Recent studies have shown that HAMLET augments the activity of antibiotics against *S. pneumoniae*, *Streptococcus pyogenes*, *Streptococcus agalactiae*, and *M. tuberculosis* [19,20,21,22]. For instance, treatment of antibiotic-resistant *S. pneumoniae*, *S. pyogenes*, and *S. agalactiae* isolates with HAMLET in combination with antibiotics (e.g., penicillin and erythromycin) reduced their Minimum Inhibitory Concentrations (MICs) [22]. These findings highlight the vast potential for the use of a novel therapeutic strategy based on HAMLET-antibiotic combination against antibiotic-resistant pathogens, which pose a menace to global public health. Furthermore, Vansarla et al. have further pointed out that HAMLET holds the ability to modulate the function of antigen-presenting cells (APCs) like dendritic cells (DCs) [23]. In vitro stimulation of primary human monocyte-derived DCs (Mo-DCs) with HAMLET not only led to the enhanced production of proinflammatory cytokines like IL-6 and IL-12, but also upregulated the surface costimulatory molecule CD83 [23]. Furthermore, HAMLET-stimulated Mo-DCs were more effective in eliciting allogeneic T cell proliferation in a mixed lymphocyte reaction (MLR) assay compared to unstimulated Mo-DCs, underscoring the HAMLET’s immunomodulatory properties [23]. However, it remains unknown whether treatment with HAMLET alone or in combination with antibiotics can alter T cell immunity against infections.

Here, we sought to determine whether amoxicillin and/or HAMLET alter T cell immunity using a combination of an infant mouse model and in vitro antigenic (killed *S. pneumoniae*) stimulation assays. We selected *S. pneumoniae* as a model organism in this study because it is a pathogen of public health significance and induces T cell responses, particularly Th17 and Th1 responses [24,25,26,27]. Our results furnish crucial information on how amoxicillin, HAMLET, or a combination of both modulates peripheral and lung Th17 and Th1 immunity to *S. pneumoniae*, which may be important for designing better therapeutic strategies.

## 2. Results

### 2.1. Treatment of Infant Mice with Amoxicillin, but Not HAMLET, Suppresses Lung IL-17A Responses to S. pneumoniae

To assess the impact of antimicrobial therapy on infant T cell responses, we treated infant mice intranasally with amoxicillin, HAMLET, or a combination of both and stimulated the lung cells and splenocytes isolated from them with UV-killed *S. pneumoniae* to measure the production pattern of T cell surrogate cytokines (IL-17A and IFN-γ). The lung cells isolated from amoxicillin- or amoxicillin plus HAMLET-treated infants produced reduced quantities of IL-17A but not IFN-γ, compared to mice receiving PBS (control) or HAMLET (Figure 1). No difference was observed with HAMLET alone compared to the control. In the case of splenocytes, none of the treatments had a significant effect on IL-17A and IFN-γ production (Figure 1). Without antigenic stimulation, the levels of IL-17A and IFN-γ produced by lung cells and splenocytes from infant mice treated with amoxicillin, HAMLET, amoxicillin plus HAMLET, or PBS did not differ statistically (Appendix A).

### 2.2. Exposure of Infants to Amoxicillin Diminishes Th17, but Not Th1, Responses

We assessed the immune responses induced by the lung and splenic CD4+ and CD8+ T cells of infant mice that received antimicrobial treatment via the intranasal route. Flow cytometric intracellular cytokine analysis demonstrated that lung CD4+, but not CD8+, T cells from amoxicillin- or HAMLET plus amoxicillin-exposed mice expressed lower levels of IL-17A than those from HAMLET alone-exposed or control mice (Figure 2 and Appendix A). In addition, there was no difference between mouse groups treated with amoxicillin and amoxicillin plus HAMLET (Figure 2). On the other hand, splenic CD4+IL-17A+ and CD8+IL-17A+ T cells did not show a significant difference between the exposed and control groups (Figure 2). Furthermore, we investigated the effect of amoxicillin and/or HAMLET on T cell responses characterized by IFN-γ production in response to *S. pneumoniae* in vitro stimulation (Figure 3). IFN-γ levels were similar in splenic and lung CD4+ and CD8+ T cells from amoxicillin- or HAMLET plus amoxicillin-exposed mice (Figure 3). Overall, these findings show that amoxicillin treatment regimens are mainly responsible for suppressing lung Th17 immunity to *S. pneumoniae*.

### 2.3. Amoxicillin Alone or with HAMLET Reduces the CD4+ T Cell Number

Neonatal exposure to antibiotics, including amoxicillin, has been shown to alter the number of T cells in the blood and the spleen [11,15]. We sought to assess whether the exposure to amoxicillin or HAMLET plus amoxicillin alters the number of T cell subsets in the spleen and lungs of infant mice. Following intranasal administration of mice with amoxicillin or HAMLET plus amoxicillin, we noticed a significant decline in the percentage and absolute number of CD4+, but not CD8+, T cells in the lungs compared to PBS-treated mice (Figure 4). Furthermore, there was no difference between groups treated with amoxicillin and amoxicillin plus HAMLET, suggesting that the suppressive effect on CD4+ T cell number was related mainly to amoxicillin (Figure 4). However, the number of CD4+ and CD8+ T cells in the spleen remained unaffected (Figure 4).

## 3. Discussion

In this study, we focused on how amoxicillin treatment of infant mice alters peripheral (splenic) and local (lung) Th17/Th1 responses to *S. pneumoniae*. We also investigated whether HAMLET, alone or with amoxicillin, impacts Th17/Th1 antipneumococcal immunity. Our main findings were that: (1) amoxicillin treatment suppressed lung IL-17A/Th17 responses to *S. pneumoniae*, but not IFN-γ/Th1 responses; (2) HAMLET treatment had no significant effect on splenic and lung Th17/Th1 immunity; and (3) amoxicillin exposure resulted in decreased CD4+ T cell numbers in the lungs. Overall, these findings provide important information on the potential impact of amoxicillin and HAMLET on infant T cell immunity to *S. pneumoniae*.

Our finding that amoxicillin suppressed lung Th17 immunity to *S. pneumoniae* is consistent with our previous findings that therapeutic regimens containing piperacillin and the β-lactamase inhibitor tazobactam reduced the frequencies of neonatal splenic and lung CD4+IL-17A+ T cells in response to *S. pneumoniae* in vitro stimulation [12]. However, we did not find any impact of amoxicillin treatment on peripheral (splenic) CD4+IL-17A+ T cells, which could be due to different routes of antibiotic administration in these studies, as well as to responses specific to the antibiotic type used or the presence/absence of beta-lactamase inhibitor. While mice received amoxicillin intranasally in this study, piperacillin plus tazobactam was injected via the intraperitoneal route [13]. It is possible that the effect of intranasal amoxicillin treatment was mainly confined to the local respiratory microbiota and T cell responses with a lower systemic exposure. Moreover, in line with the immunosuppressive role of amoxicillin, intramuscular injection of neonatal rats with meropenem and vancomycin resulted in diminished intestinal Th17 immunity to the fungus *Candida albicans* [28]. On the other hand, it remains unclear as to how antibiotic regimens alter infant T cell immunity to pathogens. Recent reports have shown that infant mice infected with vaccinia virus and exposed to antibiotics exhibited reduced frequencies of splenic DCs expressing CD11c^hi^MHC-II^hi^ [11]. The question of whether antibiotic exposure modulates DC function to generate neonatal and infant T cell immunity, including Th17, warrants further investigation. Overall, suppression of Th17 function by antibiotic regimens as shown in this study could have important implications because a Th17 response is critical to protection against neonatal and infant infections by extracellular bacterial and fungal pathogens, including *S. pneumoniae* [29].

In the present study, we chose the intranasal route of delivery for HAMLET and amoxicillin. This route was previously used to demonstrate the effect of HAMLET in combination with gentamicin in protecting mice from pneumococcal colonization [20]. The advantages of the intranasal route include ease of use and the potential to augment bioavailability and reduce adverse effects. Intranasal delivery of antibiotics faces, however, numerous challenges, particularly in relation to drug stability. Several lines of study are now being explored in an attempt to expand the range of antibiotics for intranasal delivery [30]. Amoxicillin intranasal delivery was primarily used in our study to reduce stress and disturbance in young pups by having an additional route of administration. While it showed an effect on immune responses, it is noteworthy that amoxicillin has not yet been developed for the prevention or treatment of human infections using the intranasal route.

HAMLET is not only bactericidal against certain pathogens, but also immunomodulatory [20,23]. In a mouse model of nasopharyngeal colonization with *S. pneumoniae*, intranasal administration of a combination of HAMLET and gentamicin, but not HAMLET alone, showed a significantly enhanced pneumococcal death in the nasal wash compared with the mice exposed to gentamicin alone. This indicates the ability of HAMLET to increase the efficacy of antibiotic activity in vivo [20]. In addition, using primary human immune cells in an in vitro setting, HAMLET was shown to possess immunomodulatory properties as exhibited by increased T cell proliferation by HAMLET-pulsed Mo-DCs [23]. In this study, we assessed, for the first time, the role of HAMLET in eliciting T cell immunity to *S. pneumoniae* using an infant mouse model. Our findings showed that antipneumococcal lung Th17 responses induced by intranasal HAMLET or PBS in infant mice were significantly higher than those induced by a combination of HAMLET and amoxicillin, and that there were no significant differences between Th17/Th1 responses in HAMLET- and PBS-treated mice. Thus, HAMLET exposure is neither suppressive nor stimulatory in generating Th17/Th1 responses to *S. pneumoniae* under the conditions used in this study. A positive inference drawn out of this finding is that HAMLET can potentially be used as a safe antimicrobial drug that does not suppress T cell immunity required for specific and long-lasting protection against pathogens. It is important to note that mouse infants were not weaned while receiving HAMLET or HAMLET plus amoxicillin, and it is possible that their dams’ milk may have contained HAMLET-like antimicrobials. Future studies are required to explore whether HAMLET-like antimicrobials are present in murine milk.

Collectively, our study found that exposure of infant mice to amoxicillin impairs lung Th17 responses to *S. pneumoniae*. Considering an important role for Th17 immunity in contributing to pathogen defense and the protection mediated by vaccines [25,27,31,32,33], our findings that show amoxicillin-induced suppression of Th17 responses could have important implications for the development of better therapeutic and prophylactic strategies for neonates and infants. Although the use of amoxicillin in neonatal and infantile clinical settings is appreciated, it is worth keeping in mind that treatment with amoxicillin may have suppressive effects on immune function. Amoxicillin-induced Th17 suppression can: (1) raise susceptibility to bacterial, parasitic, and viral infections; (2) change the clinical features of an infection; and (3) increase the likelihood that the live vaccine strain will develop virulence upon administration and reduce the efficacy of inactivated vaccines [34]. Additionally, unlike amoxicillin regimens, HAMLET alone did not suppress T cell immunity, accentuating its potential therapeutic role as a bactericidal without being immunosuppressive. However, caution should be taken while extrapolating our mouse data to human infants due to differences in their microbiota proportion and abundance. There are some limitations to this study. We did not evaluate the susceptibility to *S. pneumoniae* in vivo and did not explore the mechanisms of action by amoxicillin on impaired T cell responses. Future work is needed to investigate: (1) the long-term effects of amoxicillin and/or HAMLET treatment; (2) whether amoxicillin-induced Th17 immunosuppression can lead to increased susceptibility to pneumococcal infection; and (3) the underlying mechanisms by which amoxicillin exposure alters Th17 immunity.

## 4. Materials and Methods

### 4.1. Streptococcus pneumoniae

We used the *S. pneumoniae* TIGR4 strain throughout this study [35]. Pneumococcal cells were maintained in TSB (Beckton Dickinson, NJ, USA) and glycerol (15%), and kept at −80 °C. This stock culture was taken out, thawed, and grown at 37 °C to an optical density (OD) of 0.5 at 600 nm in a 5% CO_2_ incubator. Harvesting of pneumococcal cells was done by centrifugating at 5000× *g* for 10 min at 4 °C and subsequent washing in endotoxin free Dulbecco’s-PBS (Sigma-Aldrich, St. Louis, MO, USA). The pneumococcal suspension was UV-inactivated at the rate of 1200 J/m^2^ UV radiation for 30 min, aliquoted, and frozen at −80 °C for further use. The pneumococcal colonies were confirmed to be dead by culture, with the probable limit of detection being less than one in one million pneumococci.

### 4.2. HAMLET Production

Human alpha-lactalbumin was enriched from human milk and was converted into HAMLET by complexing the apo-protein (treated with EDTA to remove its calcium ion) with oleic acid (C18:1; Sigma-Aldrich) on a DEAE-containing ion exchange matrix as described [17,36]. The HAMLET complex was eluted with salt and dialyzed with water to remove salt, and the desalted protein-lipid complex was lyophilized and saved at −20 °C until use.

### 4.3. Mice

Specific pathogen free (SPF) pregnant Swiss mice were purchased from the commercial animal supplier JANVIER LABS, France. The mice were kept at the animal facility at the Oslo University Hospital, Rikshospitalet, Norway. The pregnant mice delivered pups in IVC cages, and the newborn pups stayed with their dams. Each mouse litter of 11–12 infants was taken as an experimental group. To get rid of litter bias, we randomly mixed the newborn littermates across the four experimental groups in the first few days after birth. The 16–17-day-old mouse infants in different groups were intranasally administered with HAMLET (100 μg in 10 μL PBS per pup), amoxicillin (200 μg in 10 μL PBS per pup), HAMLET plus amoxicillin (100 μg HAMLET + 200 μg amoxicillin in 10 μL PBS per pup), or PBS (10 μL per pup) daily for 7 consecutive days. The HAMLET dosage was calculated as described previously [20]. The infant mice receiving HAMLET, amoxicillin, HAMLET plus amoxicillin, or PBS were euthanized using an intraperitoneal pentobarbital injection (dose rate of 0.05–0.5 mL per mouse) under isoflurane anesthesia (4–5%). Of note, mouse experimental protocols were approved by the Norwegian Food Safety Authority, Oslo, Norway (FOTS number 21062), and the experiments were conducted in line with the institutional guidelines.

### 4.4. Cell Isolation and Antigenic Stimulation

Spleens were mashed on a 70 µm cell strainer (ThermoFisher Scientific, Rockford, IL, USA) and washed with the washing buffer (PBS, 0.5% BSA and 5 mM EDTA). The splenic cell suspension was lysed with red blood cell (RBC) lysis buffer and washed two times. On the other hand, lungs were digested in 10 mg/mL collagenase XI (Sigma-Aldrich, Israel) in RPMI 1640 supplemented with 10% heat-inactivated FBS and gentamicin (25 μg/mL) (Sigma-Aldrich, United Kingdom) for 1 h at 37 °C [37]. The lung cell suspension was treated with RBC lysis buffer (eBioscience, San Diego, CA, USA), and washed with a washing buffer containing PBS, 0.5% BSA, and 5 mM EDTA. Trypan blue was used to count live cells in the hemocytometer. In addition to evaluating cell viability by Trypan blue staining (90–96% viability), we used flow cytometric FSC versus SSC analysis to exclude debris and dead cells (Appendix A), which have low forward scatter. 2.5 × 10^6^ splenocytes in 500 μL or 5 × 10^5^ lung cells in 200 μL of complete RPMI 1640 medium having 10% heat-inactivated FBS (Sigma-Aldrich, UK) were cultured at 37 °C. The cell culture was stimulated with UV-killed *S. pneumoniae* TIGR4 (10^5^ CFU/mL) for 72 h. The culture supernatants were frozen at −80 °C, and the supernatant concentrations of IL-17A and IFN-γ were measured by Ready-SET-Go ELISA kits (eBioscience, San Diego, CA, USA) as per the manufacturer’s instructions. The cytokine detection limit of the ELISA kit for IL-17 was 4 pg/mL, whereas the limit for IFN-γ was 15 pg/mL.

### 4.5. Flow Cytometric Analysis

To perform cell surface staining, lung and splenocytes were stained with anti-CD4-Phycoerythrine (PE), anti-CD8-Flurorescein isothiocyanate (FITC), and anti-CD3-PE-Cy7 (eBioscience, San Diego, CA, USA). The isotype controls of these fluorochrome-conjugated antibodies were also used. To perform intracellular cytokine staining by flow cytometry, 2.5 × 10^6^ splenocytes in 500 μL or 5 × 10^5^ lung cells in 200 μL of complete RPMI 1640 medium with 10% heat-inactivated FBS and gentamicin (25 μg/mL) (Sigma-Aldrich, UK) were cultured at 37 °C. The cell culture was stimulated with the UV-killed *S. pneumoniae* TIGR4 (10^5^ CFU/mL) for 72 h. Following cell stimulation, splenic and lung cells were washed, cultured in the complete RPMI, and treated with a cell stimulation cocktail for 18 h (eBioscience, San Diego, CA, USA). Of note, the cell stimulation cocktail contains a mixture of phorbol 12-myristate 13-acetate (PMA), ionomycin, brefeldin A, and monensin. Brefeldin A and monensin result in the accumulation of secreted proteins in the endoplasmic reticulum and Golgi apparatus. The cells were then washed and incubated with FcR-blocking antibodies (anti 16/32; eBioscience) for 15 min. The cell surface markers (CD4, CD8, and CD3) were stained with anti-CD3-PECy7, anti-CD8-FITC, and anti-CD4-PE (eBioscience, San Diego, CA, USA). After washing, cells were treated with IC fixation buffer (Invitrogen, CA, USA), followed by permeabilization with permeabilization buffer (eBioscience, San Diego, CA, USA). Intracellular cytokine staining of cells was performed with anti-IL-17A–allophycocyanin (APC), anti-IFN-γ-APC, or isotype control antibodies (eBioscience, San Diego, CA, USA). At the end, cells were washed and resuspended in Dulbecco’s PBS mixed with 0.5% BSA and 1mM EDTA. The samples were run on a BD LSR II flow cytometer (BD Biosciences, San Diego, CA, USA) to collect experimental data. The flow cytometry data analysis was performed by the FCS Express software (De Novo Software, Los Angeles, CA, USA).

### 4.6. Statistics

One-way ANOVA and Tukey post hoc test were used for comparing four experimental groups of infant mice using GraphPad Prism Software (version 9, San Diego, CA, USA). A *p* value of less than 0.05 was considered significant.

## Figures and Tables

**Figure 1 antibiotics-12-00423-f001:**
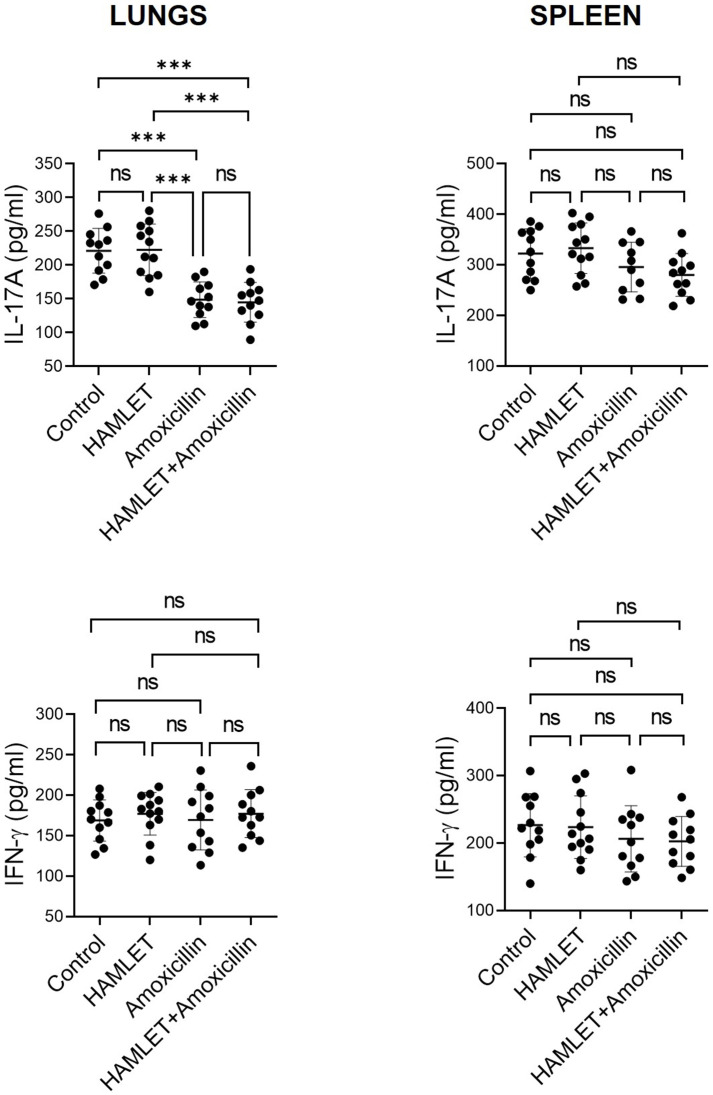
IL-17A and IFN-γ production by lung cells and splenocytes from the infant mice treated with antimicrobials. Lung cells and splenocytes from the infant mice treated with amoxicillin, HAMLET, amoxicillin plus HAMLET, or PBS (control) were stimulated with killed *S. pneumoniae* for 72 h, and Th17 (IL-17A) and Th1 (IFN-γ) cytokine levels in the culture supernatants were measured by ELISA. Each experimental group had 11–12 mice. The data are represented as the mean ± SD of two independent experiments. The dots represent the data for each mouse, and the horizontal bars are the mean values for the groups. *** *p* < 0.001. ns = non-significant. One-way ANOVA and Tukey post hoc test.

**Figure 2 antibiotics-12-00423-f002:**
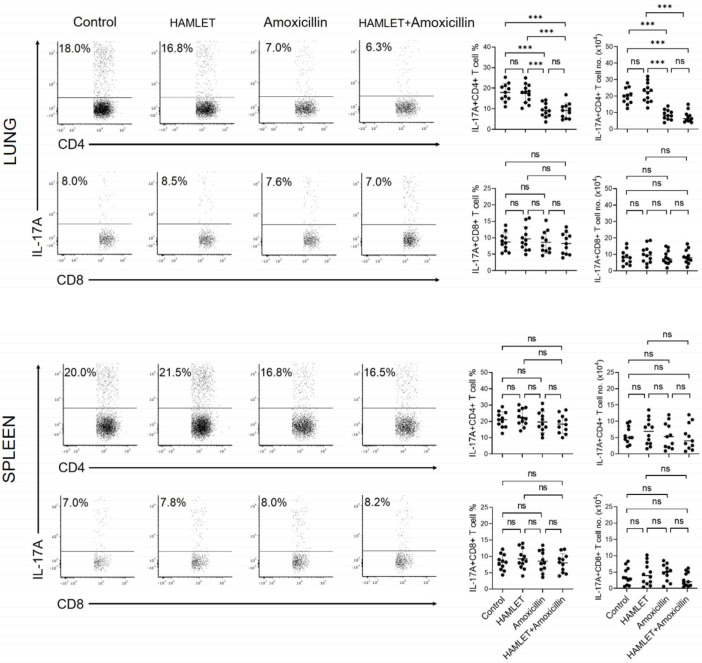
Production of IL-17A by infant CD4+ and CD8+ T cell subsets following stimulation with killed *S. pneumoniae*. The production of IL-17A by CD4+ and CD8+ T cells in the lungs and spleen was examined using flow cytometric intracellular cytokine analysis. Images of flow cytometric dot plots (**left**) and a summary of the percentages and numbers of IL-17A+ T cells (**right**). CD3+ cells were gated and presented as CD3+CD4+ and CD3+CD8+ T cells. In Appendix A, the strategies for CD4+ and CD8+ T cell gating and intracellular cytokine expression analysis are shown. There were 11–12 infant mice in each experimental group. The information on the right graph is presented as the mean ± SD and represents two independent experiments. The horizontal bars show the mean values for the groups, and the dots show data from each individual mouse. *** *p* < 0.001. ns = non-significant. One-way ANOVA and Tukey post hoc test.

**Figure 3 antibiotics-12-00423-f003:**
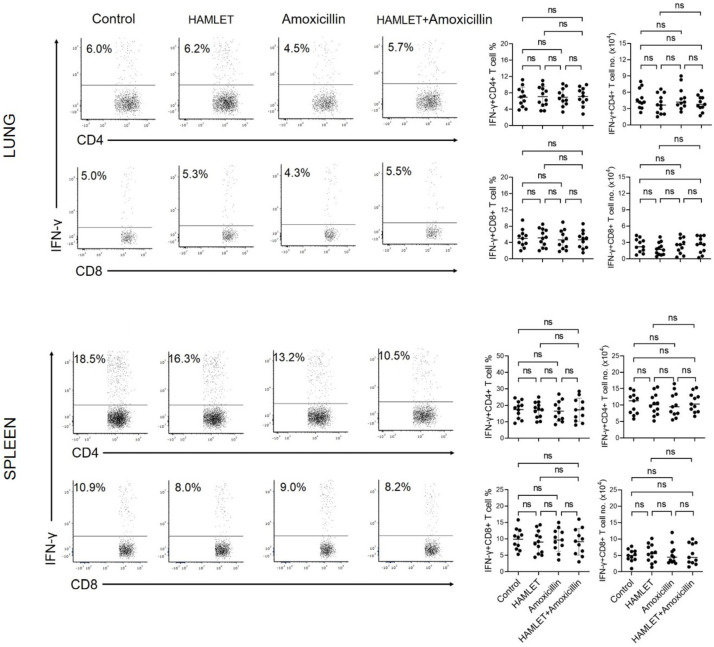
Th1 responses characterized by IFN-γ production after stimulation with killed *S. pneumoniae*. IFN-γ production by lung and splenic CD4+ and CD8+ T cells was analyzed by flow cytometric intracellular cytokine analysis. Images of flow cytometric dot plots (**left**) and a summary of the percentages and numbers of IFN-γ+ T cells (**right**). CD3+ cells were gated and presented as CD3+CD4+ and CD3+CD8+ T cells. In Appendix A, specific strategies for CD4+ and CD8+ T cell gating and intracellular cytokine expression analysis are shown. There were 11–12 infant mice in each experimental group. The information on the right graph is presented as the mean ± SD and represents two independent experiments. The horizontal bars show the mean values for the groups, and the dots show data from each individual mouse. ns = non-significant. One-way ANOVA and Tukey post hoc test.

**Figure 4 antibiotics-12-00423-f004:**
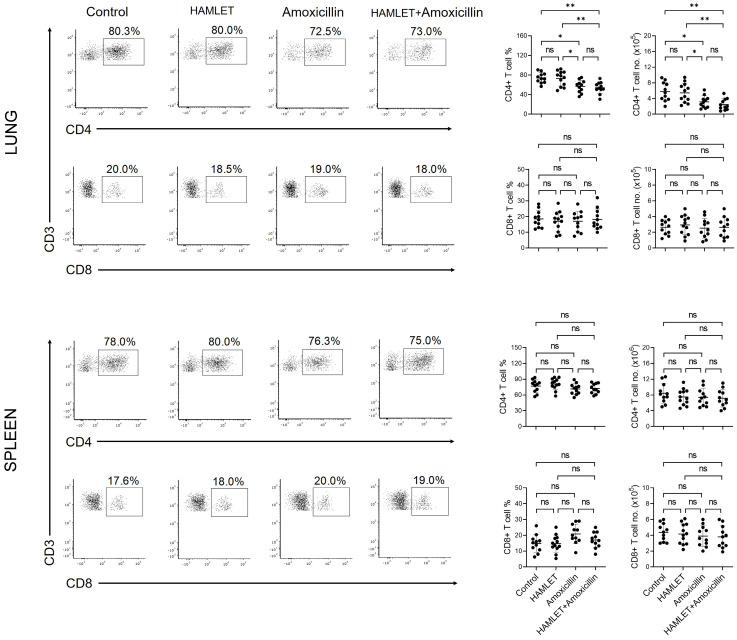
Effect of antimicrobial treatment on the percentage and number of T cells. Lung cells and splenocytes isolated from amoxicillin-, HAMLET-, or amoxicillin plus HAMLET-treated or control infant mice were stained with antibodies and analyzed by flow cytometry. The T cell subsets were gated and presented as described in Appendix A. Representative flow cytometric dot plot images (**left**) and a summary of the percentages and numbers of CD4+ and CD8+ T cells (**right**). Each experimental group had 11–12 mice. The data, which are shown as mean ± SD, represent two independent experiments. The horizontal bars represent the mean values for the groups, and each dot symbol represents data from a single mouse. ns = non-significant. * *p* < 0.05; ** *p* < 0.01. One-way ANOVA and Tukey post hoc test.

## Data Availability

We confirm that the data supporting the findings of this study are available within the manuscript and its Appendix A.

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
