# Peer review of "Treatment of Mouse Infants with Amoxicillin, but Not the Human Milk-Derived Antimicrobial HAMLET, Impairs Lung Th17 Responses"

_antibiotics, 2023, doi:10.3390/antibiotics12020423_

Round 1
Reviewer 1 Report
Shekhar et al perform a well designed study on the immunomodulatory effects of the antibiotic amoxicillin and potential synergies with the human derived antimicrobial HAMLET. While overall a strong study and strong report, there are several instances in the text which require more clarification.
Major concern: the route of administration of antibiotic and HAMLET is via this intranasal route. This is not biologically relevant, as these compounds would be administered orally, or in the case of amoxicillin, potentially intravenously. This could explain the reactivity only being in the lung associated T cells and not the splenocytes. While new experiments would be preferred this reviewer would accept a better clarification in the text as to the limitations that the study design provides and lack of relevance to the human infant system.
Minor concerns:
Line 34-36: the current wording suggests that antibiotics inflict severe morbidity and mortality, while the authors probably intend to suggest that sepsis does.
Figure 1: can the authors provide data from cells from animals treated with HAMLET and or amoxicillin in the absence of heat killed pneumococcal stimulation (or a note in the text if there is no stimulation/ no difference in stimulation in the absence of bacterial signals)
Figures 2 and 3: is this done in the context of heat killed pneumococcal stimulation?
Lines 277-292: were the mice weaned at the time of treatment, if not how do you account for lactational derived antimicrobial peptides that could be acting similar to HAMLET?
Lines 305-306 and 317-318: Why was a different MOI used for lung cells versus splenocytes?
Line 308: as amoxicillin is a lytic antibiotic, would not lysed bacterial cells be a more biologically relevant stimulation signal?
Author Response
REVIEWER # 1
Shekhar et al perform a well-designed study on the immunomodulatory effects of the antibiotic amoxicillin and potential synergies with the human derived antimicrobial HAMLET. While overall a strong study and strong report, there are several instances in the text which require more clarification.
Major concern: the route of administration of antibiotic and HAMLET is via this intranasal route. This is not biologically relevant, as these compounds would be administered orally, or in the case of amoxicillin, potentially intravenously. This could explain the reactivity only being in the lung associated T cells and not the splenocytes. While new experiments would be preferred this reviewer would accept a better clarification in the text as to the limitations that the study design provides and lack of relevance to the human infant system.
Response: Thank you for raising this important issue. In this study, we chose the intranasal route of delivery for HAMLET and amoxicillin. This route was previously used to demonstrate the protective effect of HAMLET in combination with gentamicin in protecting mice from pneumococcal colonization (Marks et al., PLOS ONE, 2012). The advantages of the intranasal route include ease of use and the potential to augment bioavailability and reduce adverse effects. Intranasal delivery of antibiotics faces, however, numerous challenges, particularly in relation to drug stability. Several lines of study are now being explored in an attempt to expand the range of antibiotics for intranasal delivery (Mardikasari et al., J Drug Delivery Sci Tech, 2022). Amoxicillin intranasal delivery was primarily used in our study to reduce stress and disturbance in young pups by having an additional route of administration. While it showed an effect on immune responses, it is noteworthy that amoxicillin has not yet been developed for the prevention or treatment of human infections using the intranasal route. We have amended the manuscript to address this issue in the discussion (Line # 243-254).
Minor concerns:
- Line 34-36: the current wording suggests that antibiotics inflict severe morbidity and mortality, while the authors probably intend to suggest that sepsis does.
Response: This sentence has been corrected (Line # 35).
- Figure 1: can the authors provide data from cells from animals treated with HAMLET and or amoxicillin in the absence of heat killed pneumococcal stimulation (or a note in the text if there is no stimulation/ no difference in stimulation in the absence of bacterial signals)
Response: The data without bacterial stimulation have been included as Supplementary figure 2.
- Figures 2 and 3: is this done in the context of heat killed pneumococcal stimulation?
Response: The cells were stimulated with the UV-killed S. pneumoniae. We have modified the Figure legends 2 and 3 to include this information.
- Lines 277-292: were the mice weaned at the time of treatment, if not how do you account for lactational derived antimicrobial peptides that could be acting similar to HAMLET?
Response: Thank you for this interesting and pertinent question. The mice were not weaned during treatment with HAMLET and/or amoxicillin. It's possible that murine milk contains HAMLET-like antimicrobials. In this study, there was, however, no discernible difference between the animal groups treated with and without HAMLET, and the differences we did find were only present in mice exposed amoxicillin.
- Lines 305-306 and 317-318: Why was a different MOI used for lung cells versus splenocytes?
Response: We had to use 5 x 105 lung cells in 200 μl of complete RPMI medium as opposed to 2.5 x 106 splenocytes in 500 μl of medium due to the scarce number of lung cells in mouse pups. Nevertheless, for the in vitro assays, the same lung cell concentration was utilised across all treatment groups.
- Line 308: as amoxicillin is a lytic antibiotic, would not lysed bacterial cells be a more biologically relevant stimulation signal?
Response: In mouse pups receiving amoxicillin treatment, lysed pneumococcal cells might be more biologically relevant stimulants. However, we used the intact UV-killed S. pneumoniae to measure immune responses because stimulation with disintegrated S. pneumoniae results in significantly reduced immune responses in vitro (IFN-γ, TNF, and IL-12) as compared to intact bacteria (Martner et al., Infect Immun, 2009; https://www.ncbi.nlm.nih.gov/pmc/articles/PMC2738010/).
Reviewer 2 Report
The authors investigated the immune response to HAMLET and/or amoxicillin upon S. pneumoniae infection. They revealed an interesting aspect of amoxicillin treatment showing immunomodulatory effects on Th17/Th1 responses. The authors also checked whether HAMLET can change these responses The manuscript is well-written, structured and easy-to follow. The concept makes sense and is in line with previous studies.
Major concerns:
- - The authors treated mice intranasal amoxicillin/ HAMLET or both then isolated their lung and spleen cells. Why they didn’t infected mice with live S. pneumoniae to set up a complete mice model?
- - Did HAMLET or amoxicillin affect Th17 response in vivo alone? Did the authors measured IL17A or IFN-g in BAL or serum before cell isolation?
- - Based on flow cytometry, in lung CD4+ T cells the IL-17 expression was decreased by amoxicillin. What can be the pathomechanism behind this? Did the authors check other markers with flow cytometer associated with IL-17A signalling ( such as IL17R etc.)?
- - IL-17A drives other cytokines such as GMCSF. Did the authors check the effects of decreased IL17 production on other cytokines? If not, why not?
- - How IL17A gene expression changed in mice lung and spleen?
- - Besides T cell, did the authors experience a change in neutrophil cell number/ granularity by flow cytometer? It would be better to show data about other cells too from the treated, digested mice lung? Can HAMLET with and without amoxicillin, or amoxicillin alone influence other cells?
Minor concerns:
- - On Figure 2 and 3, the absolute cell number data would be also helpful to understand flow data.
- - On Figure 2, a histogram ( MFI normalized to mode) would be good to include and compare the expression of IL17A in CD4+ and CD8+ T cells.
- - IL-17A has important role in mucosal immunity and defense against several bacteria and parasites. What can be the clinical relevance of this research? It would be nice to include the potential benefits in the discussion section.
Author Response
REVIEWER # 2
The authors investigated the immune response to HAMLET and/or amoxicillin upon S. pneumoniae infection. They revealed an interesting aspect of amoxicillin treatment showing immunomodulatory effects on Th17/Th1 responses. The authors also checked whether HAMLET can change these responses. The manuscript is well-written, structured and easy-to follow. The concept makes sense and is in line with previous studies.
Major concerns:
- The authors treated mice intranasal amoxicillin/ HAMLET or both then isolated their lung and spleen cells. Why they didn’t infected mice with live S. pneumoniae to set up a complete mice model?
Response: Thank you for this important question. While designing the present study in the latter half of 2021, we attempted to optimize the pneumococcal infection model in mouse pups in our infection unit settings, but failed to do so due to working restrictions inflicted by the COVID-19 pandemic. Thereafter, we decided to go with the experimental design without involving infections. Of note, we plan to treat mouse pups with HAMLET and/or amoxicillin preceded or followed by a pneumococcal challenge in future studies.
- Did HAMLET or amoxicillin affect Th17 response in vivo alone? Did the authors measured IL17A or IFN-g in BAL or serum before cell isolation?
Response: We did not measure Th1 and Th17 responses in vivo (BAL and serum) because of our focus on antigen-stimulated T cell responses. We attempted to collect nasal wash and BAL from mouse pups but were unable to do so due to the pups' tiny tracheal tubes. Of note, we have data on the in vitro cytokine production of lung and splenic cells without pneumococcal stimulation, which are included in Supplementary figure 2.
- Based on flow cytometry, in lung CD4+ T cells the IL-17 expression was decreased by amoxicillin. What can be the patho-mechanism behind this? Did the authors check other markers with flow cytometer associated with IL-17A signalling (such as IL17R etc.)?
Response: The mechanistic details of how antibiotics alter IL-17 expression in CD4+ T cells are largely unclear. A recent study has shown that the antibiotic Linezolid disrupts IL-17 production in Th17 cells by targeting mitochondrial translation (Almeida et al., Immunity, 2021). Whether it is true in the case of amoxicillin remains to be explored. In this study, we were confined to measuring IL-17A signaling due to its crucial role in antipneumococcal immunity.
- IL-17A drives other cytokines such as GMCSF. Did the authors check the effects of decreased IL17 production on other cytokines? If not, why not?
Response: No. This was beyond the purview of the present study.
- How IL17A gene expression changed in mice lung and spleen?
Response: We did not investigate the IL-17A gene expression in murine lung and spleen.
- Besides T cell, did the authors experience a change in neutrophil cell number/ granularity by flow cytometer? It would be better to show data about other cells too from the treated, digested mice lung? Can HAMLET with and without amoxicillin, or amoxicillin alone influence other cells?
Response: Since our objective was to study the impact of amoxicillin and HAMLET on T cells, we analyzed the frequencies and function of major T cell subsets – CD4+ and CD8+ T cells. HAMLET and/or amoxicillin can influence the other cell types. Exposure of neonatal animals to antibiotics, including amoxicillin, have been shown to alter the number of T cells in the blood and the spleen (Gonzalez-Perez et al., J Immunol, 2016; Fouhse et al., Front Immunol, 2019). Furthermore, although HAMLET activated dendritic cells and macrophages in vitro (Vansarla et al., Eur J Immunol, 2021), it is yet to be explored whether HAMLET modulates the other immune cells in vivo.
Minor concerns:
- On Figure 2 and 3, the absolute cell number data would be also helpful to understand flow data.
Response: The absolute number data have been included in the Figure 2 and 3.
- On Figure 2, a histogram (MFI normalized to mode) would be good to include and compare the expression of IL17A in CD4+ and CD8+ T cells.
Response: We have shown the histogram plots for IL-17A expression in the lung and splenic CD4+ and CD8+ T cells and a summary graph of the MFI of the IL-17 expression in T cells (Supplementary figure 3).
- IL-17A has important role in mucosal immunity and defense against several bacteria and parasites. What can be the clinical relevance of this research? It would be nice to include the potential benefits in the discussion section.
Response: We have modified the manuscript to include a brief discussion on the clinical relevance of this research (Lines # 285-288).